# Semiotic Aggregation in Deep Learning

**DOI:** 10.3390/e22121365

**Published:** 2020-12-03

**Authors:** Bogdan Muşat, Răzvan Andonie

**Affiliations:** 1Department of Electronics and Computers, Transilvania University, 500036 Braşov, Romania; 2Xperi Corporation, 3025 Orchard Parkway, San Jose, CA 95134, USA; 3Department of Computer Science, Central Washington University, Ellensburg, WA 98926, USA; razvan.andonie@cwu.edu

**Keywords:** deep learning, spatial entropy, saliency maps, semiotics, convolutional neural networks

## Abstract

Convolutional neural networks utilize a hierarchy of neural network layers. The statistical aspects of information concentration in successive layers can bring an insight into the feature abstraction process. We analyze the saliency maps of these layers from the perspective of semiotics, also known as the study of signs and sign-using behavior. In computational semiotics, this aggregation operation (known as superization) is accompanied by a decrease of spatial entropy: signs are aggregated into supersign. Using spatial entropy, we compute the information content of the saliency maps and study the superization processes which take place between successive layers of the network. In our experiments, we visualize the superization process and show how the obtained knowledge can be used to explain the neural decision model. In addition, we attempt to optimize the architecture of the neural model employing a semiotic greedy technique. To the extent of our knowledge, this is the first application of computational semiotics in the analysis and interpretation of deep neural networks.

## 1. Introduction

Convolutional neural networks (CNNs) were first made popular by Lecun et al. [1] with their seminal work on handwritten character recognition, where they introduced the currently popular LeNet-5 architecture. At that time, computational power constraints and lack of data prohibited those CNNs from achieving their true potential in terms of computer vision capabilities. Years later, Krizhevsky et al. [2] marked the start of the current deep learning revolution, when, during the ILSVRC 2012 competition, their CNN, entitled AlexNet, overrun its competitor from the previous year by a margin of almost 10%. Since then, research on novel CNN architectures became very popular producing candidates like VGG [3], GoogleNet [4], ResNet [5], and more recently EfficientNet [6].

Despite the ability of generating human-alike predictions, CNNs still lack a major component: interpretability. Neural networks in general are known for their black-box type of behavior, being capable of capturing semantic information using numerical computations and gradient-based learning, but hiding the inner working mechanisms of reasoning. However, reasoning is of crucial importance for areas like medicine, law, and finance, where most decisions need to come along with good explanations for taking one particular action in favor of another. Usually, there is a trade-off between accuracy and interpretability. For instance, extracted IF-THEN rules from a neural network are highly interpretable but less accurate.

Since the emergence of deep learning, there have been efforts to analyze the interpretability issue and come up with potential solutions that might equip neural networks with a sense of causality [7,8,9,10,11,12,13]. The high complexity of deep models makes these models hard to interpret. It is not feasible to extract (and interpret) classical IF-THEN rules from a ResNet with over 200 layers.

We need different interpretation methods for deep models and an idea comes from image processing/understanding. A common technique for understanding the decisions of image classification systems is to find regions of an input image that were particularly influential to the final classification. This technique is known under various names: sensitivity map, saliency map, or pixel attribution map. We will use the term *saliency map*. Saliency maps have long been present and used in image recognition. Essentially, a saliency map is a 2D topological map that indicates visual attention priorities. Applications of saliency maps include image segmentation, object detection, image re-targeting, image/video compression, and advertising design [13].

Recently, saliency maps became a popular tool for gaining insight into deep learning. In this case, saliency maps are typically rendered as heatmaps of neural layers, where “hotness” corresponds to regions that have a big impact on the model’s final decision. We illustrate with an intuitive gradient-based approach, the Vanilla Gradient algorithm [8], which proceeds as follows: forward pass with data, backward pass to the input layer to get the gradient, and render the gradient as a normalized heatmap.

Certainly, saliency maps are not the universal tool for interpreting neural models. They focus on the input and may neglect to explain how the model makes decisions. It is possible that saliency maps are extremely similar for very different output predictions of the neural model. An example was provided by Alvin Wan (https://bair.berkeley.edu/blog/2020/04/23/decisions/#fn:saliency) using the Grad-CAM (Gradient-weighted Class Activation Mapping) saliency map generator [7]. In addition, some widely deployed saliency methods are incapable of supporting tasks that require explanations that are faithful to the model or the data generating process. Relying only on visual assessment of the saliency maps can be misleading and two tests for assessing the scope and quality of explanation methods were introduced in [14].

A good visual interpretation should be class-discriminative (i.e., localize the category in the image) and high-resolution (i.e., capture fine-grained details) [7]. Guided Grad-CAM [7] is an example of a visualization which is both high-resolution and class-discriminative: important regions of the image which correspond to any decision of interest are visualized in high-resolution detail even if the image contains evidence for multiple possible concepts.

In our approach, we focus on the statistical aspects of the information concentration processes which appear in the saliency maps of successive CNN layers. We analyze the saliency maps of these layers from the perspective of semiotics. In computational semiotics, this aggregation operation (known as superization) is accompanied by a decrease of spatial entropy: signs are aggregated into supersigns. A saliency map aggregates information from the previous layer of the network. In computational semiotics, this aggregation operation is known as *superization*, and it can be measured by a decrease of spatial entropy. In this case, signs are synthesized into supersigns.

Our contribution is an original, and to our knowledge, the first application application of computational semiotics in the analysis and interpretation of deep neural networks. *Semiotics* is known as the study of signs and sign-using behavior. According to [15], *computational semiotics* is an interdisciplinary field which proposes a new kind of approach to intelligent systems, where an explicit account for the notion of sign is prominent. In our work, the definition of computational semiotics refers to the application of semiotics to artificial intelligence. We put the notion of sign from semiotics into service to give a new interpretation of deep learning, and this is new. We use computational semiotics’ concepts to explain decision processes in CNN models. We also study the possibility of applying semiotic tools to optimize the architecture of deep learning neural networks. Currently, model architecture optimization is a hot research topic in machine learning.

The inputs for our model are saliency maps, generated for each CNN layer by Grad-CAM, which currently is a state-of-the-art method. We compute the entropy of the saliency maps, meaning that we quantify the information content of these maps. This allows us to study the superization processes which take place between successive layers of the network. In our experiments, we show how the obtained knowledge can be used to explain the neural decision model. In addition, we attempt to optimize the architecture of the neural model employing a semiotic greedy technique.

The paper proceeds as follows: Section 2 describes the visualization of CNN networks through saliency maps, with a focus on the Grad-CAM method used in our approach. Image spatial entropy and its connection to saliency maps are presented in Section 3. Section 4 introduces semiotic aggregation in the context of deep learning. Section 5 concentrates the conceptual core of or contribution—the links between semiotic aggregation and CNN saliency maps. The experimental results are described in Section 6. Section 7 discusses how semiotic aggregation could be used to optimize the architecture of a CNN. Section 8 contains final remarks and open problems.

## 2. Saliency Maps in CNNs

This section describes the most recent techniques used for the visualization of CNN layers. Overviews of saliency models applied to deep learning networks can be found in [7,13].

One of the earliest works belong to Zeiler et al. [9]. They used a Deconvolutional Network (deconvnet) to visualize the relevant parts from an input image that excite the top nine most activated neurons from a feature map (i.e., the output of a convolutional layer, a maxpool layer, a nonlinear activation function) resulted at a particular layer. A deconvnet represents a map from the hidden neuron activities back to the input pixel space by means of inverting the operations done in a CNN: unpooling, rectifying (using ReLU), and convolutional filtering. Their visualization technique strengthened the intuition that convolutional filters do indeed learn hierarchical features, starting from simple strokes and edges to object parts and in the end composing whole objects, as the depth of the network increases. Springenberg et al. [10] demonstrated that by slightly changing the way the gradient through the ReLU nonlinearity is computed—by discarding negative values (guided backpropagation)—they can visualize convolutional filters for a CNN with strided convolution instead of pooling.

In visual recognition, a saliency map (e.g., Figure 1) can capture the most important or salient features (pixels) of an input image which are responsible for a particular decision. In the case of a CNN classifier, this decision translates into finding the class with the maximum likelihood score. As can be seen in Figure 1, the saliency map can be represented as a heatmap, where the intensity represents the importance of the features.

The notion of saliency map is not novel and has been used even before the emergence of CNNs [16]. In the context of CNNs, the work of Simonyan et al. [8] was among the first ones to explore saliency maps by using as a signal the backpropagated gradients with respect to the input image. Higher magnitudes of the gradient tensor corresponded to higher importance of the respective pixels. In deep CNNs, the class-aware gradient signal is mostly lost while moving backwards to the input of the network. Thus, for a lot of images, the resulted saliency maps in [16] were noisy and difficult to interpret. The same paper introduced a method for generating class specific images: apply gradient descent on a random input noise image until convergence, in order to maximize the likelihood of a class. The resulting images managed to capture some of the semantics of a real image belonging to that class.

A related approach is SmoothGrad, proposed by Smilkov et al. [11], where the gradient corresponding to an input image is computed as the average of the gradients of multiple samples obtained by adding Gaussian noise to the original input image. This has the effect of smoothing the resulted gradient with a Gaussian kernel by means of a stochastic approximation, resulting in a less noisy saliency map.

Grad-CAM, a recent popular technique for saliency map visualization [7], uses the gradient information obtained from backpropagating the error signal from the loss function with respect to a specific feature map $A^{\left(l\right)}\in R^{w\times h\times c}$, at any layer *l* of the network, where w,h and *c* represent the width, height, and number of channels of that feature map, respectively. The gradient signal is averaged over the spatial dimensions w×h to obtain a *c*-dimensional vector of importance weights $\alpha_{k}$. The importance weights are used to perform a weighted channel-wise combination with the feature maps $A_{k}^{\left(l\right)}$ and ultimately passed through a ReLU activation function:(1)$$O_{Grad-CAM}^{\left(l\right)}=ReLU\left(\sum_{k=0}^{c}\alpha_{k}A_{k}^{\left(l\right)}\right)$$

$O_{Grad-CAM}^{\left(l\right)}$ in Formula (Equation 1) is the output resulted by applying the Grad-CAM technique on a particular layer *l*. By normalizing the values between [0,1] using the min-max normalization scheme and then, multiplying by 255, it will result in a map of pixel intensities g∈[0,255], where 255 denotes maximum importance and 0 denotes no importance.

The ReLU activation function is applied because only features that have a positive influence on the class of interest usually matter. Negative values are features likely to belong to other categories in the image. In [7], the authors justify that, without the ReLU function, the saliency maps could sometimes highlight more than just the desired class of interest.

Grad-CAM can be used to explain activations in any layer of a deep network. In [7], it was applied only to the final layer, in order to interpret the output layer decisions. In our experiments, we use Grad-CAM to generate the saliency maps of all CNN layers.

## 3. Image Spatial Entropy

Our work analyzes the entropy variations of 2D saliency maps. For this, we need to compute the entropy of 2D structures. This is very different than the approach in [16], where saliency maps are obtained from local entropy calculation. Rather than generating maps using an entropy measure, we compute the entropy of saliency maps generated by the gradient method in Grad-CAM.

The most trivial solution is to use the univariate entropy, which assumes all pixels as being independent and does not take into consideration the contextual aspect information.

A more accurate model is the Spatial Disorder Entropy (SDE) [17], which considers an entropy measure for each possible spatial distance in an image. Let us define the joint probability of pixels at spatial locations (i,j) and (i+k,j+l) to take the value *g*, respectively $g^{\prime }$:(2)$$p_{gg^{\prime }}\left(k,l\right)=P\left(X_{i,\;j}=g,X_{i+k,\;j+l}=g^{\prime }\right)$$
where *g* and $g^{\prime }$ are pixel intensity values (0−255). If we assume that $p_{gg^{\prime }}$ is independent of (i,j) (the homogeneity assumption [17]), we define for each pair (k,l) the entropy
(3)$$H\left(k,\;l\right)=-\sum_{g}\sum_{g^{\prime }}p_{gg^{\prime }}\left(k,\;l\right)\log p_{gg^{\prime }}\left(k,\;l\right)$$
where the summations are over the number of outcome values (256 in our case). A standardized relative measure of bivariate entropy is [17]:(4)$$H_{R}\left(k,\;l\right)=\frac{H(k,\;l)-H(0)}{H(0)}\in \left[0,\;1\right]$$

The maximum entropy $H_{R}\left(k,\;l\right)=1$ corresponds to the case of two independent variables. H(0) is the univariate entropy, which assumes all pixels as being independent, and we have H(k,l)≥H(0).

Based on the relative entropy for (k,l), the SDE for an m×n image X was defined in [17] as:(5)$$H_{SDE}\left(X\right)\approx \frac{1}{mn}\sum_{i=1}^{m}\sum_{j=1}^{n}\sum_{k=1}^{m}\sum_{l=1}^{n}H_{R}\left(i-k,\;j-l\right)$$

For k,l>>1, $H_{R}\left(k,\;l\right)$ is always equal or very close to one. Consequently, $H_{SDE}$ is usually very close to one (the max value) for most images, which is not convenient for our purposes. In addition, the complexity of SDE computation is high.

For these reasons, we decided to use a simplified version—the Aura Matrix Entropy (AME, see [18]), which only considers the second order neighbors from the SDE computation:(6)$$\begin{array}{c} \begin{array}{c} H_{AME}\left(X\right)\approx \frac{1}{4}\left(H_{R}\left(-1,\;0\right)+H_{R}\left(0,\;-1\right)+H_{R}\left(1,\;0\right)+H_{R}\left(0,\;1\right)\right) \end{array} \end{array}$$

The additional assumption is that the image is isotropic, which causes different orientations of the neighboring pixels to have the same entropy. In other words, the joint pdf of the vertical and horizontal neighboring process is averaged to obtain a global joint pdf of the image. This averaging makes the resulting pdf smoother and more equally distributed throughout the entire sample space. In a comparison study [19], the AME measure provided the most effective outcome among several other image spatial entropy definitions, even if it overestimates the image information.

Putting it all together, starting from a map obtained by Formula (Equation 1), we compute the probabilities $p_{gg^{\prime }}$ in Formula (Equation 2), and finally the AME in Formula (Equation 6).

## 4. Semiotic Aggregation in Deep Learning

We aim to introduce in this section the semiotic framework used to analyze visual representations (saliency maps) of multi-layered neural networks. Our main operation is aggregation, applied layer-wise in such networks. The basic computational tool is information theory, but the aggregation operation is applied in a semiotic framework and this makes our contribution interdisciplinary.

In semiotics (or *semiosis*), a *sign* is anything that communicates a meaning that is not the sign itself, to the interpreter of the sign. This definition is very general. Alternative in-depth definitions can be found in [20,21,22]. We consider the triadic model of semiosis, as stated by Charles Sanders Peirce. Peirce defined semiosis as an irreducible triadic relation between Sign–Object–Interpretant [23].

Charles Morris [24] defined semiotics as grouped into three branches:Syntactics: relations among or between signs in formal structures without regard to meaning.Semantics: relation between signs and the things to which they refer: their signified denotata, or meaning.Pragmatics: relations between the sign system and its human (or animal) user.

In a simplistic manner, semiotics already played some role in computer science during the sixties. The distinction of syntactics, semantics, and pragmatics by Charles Morris was at that time imported into programming language theory [25]. More recent results can be found in [26].

Computational semiotics is built upon a mathematical description of concepts from classic semiotics. In [27], it was stated that semantic networks can implement computational intelligence models: fuzzy systems, neural networks, and evolutionary computation algorithms. Later, some computational model of Peirce’s triadic notion of meaning processes were proposed [15,28,29].

Taking advantage of Peircean semiotics and recent results in cognitive science, Baxter et al. proposed a unified framework for the interpretation of medical image segmentation as a sign exchange in which each sign acts as an interface metaphor [30]. This framework provides a unified approach to the understanding and development of medical image segmentation interfaces. A complete computational model of Peirce’s semiosis is very complex and still not available.

According to Mihai Nadin, almost all inference engines deployed today in machine learning encode semiotic elements, although, at times, those who designed them are rather driven by semiotic intuition than by semiotic knowledge [31,32].

Recently, there is a huge interest in self-explaining machine learning models. This can be regarded as exposure of the self-interpretation and semiotic awareness mechanism. The concept of sign and semiotics offers a very promising and tempting conceptual basis to machine learning.

In this work, we focus on computational aspects of semiotics in deep learning. Our semiotic infrastructure is at the intersection of Peirce’s theory and information theory, a theory developed by Max Bense [33] and Helmar Frank [34].

The usual signs designate material entities which are unconsciously perceived. These so-called *first level signs* may be agglomerated into signs at the next hierarchical level, called *second level supersigns*. Iterating the process, we obtain more abstract *k-th level supersigns*. The transition from *k*-th level to (k+1)-th level supersigns is called *superization*. Frank [34] identified two types of superization:**Type I** “Durch Klassesbildung” (by class formation, in German): building equivalence classes and thus reducing the number of signs. The letters of a text may be considered first level signs. The equivalence class of all types of letter “a” (handwritten, capital, and so on) is a second level supersign.**Type II** “Durch Komplexbildung” (by compound formation, in German): building compound supersigns from simpler component supersigns. Reconsidering the previous example, we may obtain this way words from letters, sentences from words, and more and more complex and abstract syntactic-semantic structures afterward.

Superization is a semiotic aggregation process characterized at each perception level by a specific repertory of supersigns. Hierarchical computer vision data structures (e.g., quadtrees, multi-resolution pyramids) may be considered simplistic superizations [35,36]. The basic idea is to treat each component as a pixel at the given hierarchical level. In this case, there is a similarity between hierarchical aggregative representation and superization processes. However, there are also differences: superizations are not simple combinatorial processes, but subtle syntactic-semantic perception frames related to Peirce’s triadic model of semiosis.

A multi-resolution image representation can be characterized at each level by an information measure. The resolution-dependent Shannon entropy can be derived from the probability distribution of grey-level events observed at that level [37]. Using the newspaper’s reading analogy, at the magnified level, where only white and black patches are visible, the entropy *H* will be low. As the picture is brought to normal focusing distance, a great variety of grey levels become apparent, and, consequently, the entropy increases. As the picture is moved further away from the eyes, the entropy decreases. Finally, it may become nearly uniformly grey in appearance, with H≈0. The observation that associates with the peak value of the entropy is one of the most meaningful observations of the picture. However, because of other factors, the maximum entropy is not always associated with the “optimal” resolution [36].

From an informational psychology view, the entropy increases until it reaches its peak value. In our opinion [35,36], this phase may be associated with the informational adaptation of the perceiver. The subsequent entropy decrease is related to the processing of structural information [37]. The rate of decrease depends largely upon the amount of structural information in the picture. The entropy falls quickly when little structural information is available, whereas, when major structural information is present, the entropy will remain high over most of its range. The variation of entropy can indicate the type and quantity of structural information in the picture in terms of size and relationships to detailed features. In the current study, we focus only on the entropy decrease phase, since the analyzed CNNs do not adapt to the inputs by changing dynamically the input image resolution.

The idea of considering the CNN layers as multi-resolution representations of the input images is interesting, but not very new [38,39,40]. For instance, in [38] a spatial pyramid pooling layer is introduced between convolutional layers and fully connected layers to avoid the need for cropping or warping of the input images. In [40], the incoming convolution layers at multiple sampling rates are applied to the convolutional layers to capture objects as well as image context at multiple scales.

In our approach, we consider the multi-resolution image representation example in the context of a semiotic recognition process, where the machine (or the interpretant) attempts to classify an input image. We imagine the recognition process as a feedforward multi-layer neural classifier where each layer performs a superization of the previous layer. We assume that the subjective information (measured by the entropy) is made available to an interpretant (i.e., the computer or the human supervisor) who attempts to classify the input image.

Let us consider the entropies computed at two successive layers: $H_{k}$ and $H_{k+1}$. The extracted information by the interpretant can be measured by the difference $H_{k}-H_{k+1}$. Details can be found in [41]. We have the following result:

**Theorem** **1**(from [34])**.**
*: Superization tends to concentrate information by decreasing entropy.*

**Proof of Theorem 1.** We consider separately the two types of superization. For a set $Z=(Z_{1},\cdots ,\;Z_{n})$ of supersigns with the corresponding probabilities $p_{1},\cdots ,\;p_{n}$, $\sum p_{i}=1$, using a superization of the first type, we may obtain supersigns of the next level $Z^{*}=\left(Z_{1},\cdots ,\;Z_{n-2},\;\left\{Z_{n-1},\;Z_{n}\right\}\right)$ with the corresponding probabilities $p_{1},\cdots ,\;p_{n-2},\;p_{n-1}+p_{n}$. We have the following inequality: $H\left(Z\right)=\sum p_{i}\;log\;p_{i}\geq H\left(Z^{*}\right)$.For two sets of supersigns *X* and *Y*, using the second type of superization, we obtain compound supersigns from the joint set Z=(X,Y). A well-known relation completes the proof: H(X)+H(Y)≥H(Z). □

An intuitive application of this theorem is when we consider the neural layers of a CNN. A type I superization appears when we reduce the spatial resolution of a layer k+1 by subsampling layer *k*. This is similar to class formation because we reduce the variation of the input values (i.e., we reduce the number of signs). In CNNs, this is typically performed by a pooling operator. The pooling operator can be considered as a form of nonlinear down-sampling which partitions the input image into a set of non-overlapping rectangles and, for each such sub-region, it computes its mean (average pooling) or max value (max pooling). The formula for max pooling applied to a feature map *F* at layer *k* and locations (i,j) with a kernel of 2 × 2 is:(7)$$O_{i,j}\left(F\right)=max\left(F_{i,j},F_{i+1,j},F_{i,j+1},F_{i+1,j+1}\right)$$

A type II superization is produced when applying a convolutional operator to a neural layer *k*. As an effect, layer k+1 will focus on more complex objects, composed of objects already detected by layer *k*. The convolutional operator for a feature map *F* at layer *k* and pixel locations (i,j) with a 3 × 3 kernel *W* has the following formula:(8)$$O_{i,j}\left(F\right)=\sum_{x=0}^{2}\sum_{y=0}^{2}F\left(i+x,i+y\right)W\left(x,y\right)$$

The output *O* of the convolutional operator is a linear combination of the input features and the learned kernel weights. Thus, a resulting neuron will be able to detect a combination of simpler object forming a more complex one, by a composition of supersigns.

We observe that the effect of superization is a tendency of entropy decrease at each level. This is different than in the case of multi-resolution image representation. In [36], we explained this difference by the following thesis: “The first level signs are perceived at a complexity level which corresponds to the “optimal” resolution.” However, this thesis does not apply to a computer recognition model (a classifier), but to human perception.

In a simplified form, a multi-layered classifier can be interpreted from Morris’ semiotic theory as a transition: syntactics–semantics–pragmatics. At the end of a successful recognition process, the entropy of the output layer becomes 0 and no further information needs to be extracted. The last layer (the fully connected layer in a CNN network) is connected to the outer world, the world of objects. This may be considered the pragmatic level in Morris’ semiotic theory, since it shows the relation between the input signs and the output objects which can be related to decisions and actions.

## 5. Signs and Supersigns in CNN Saliency Maps

Theorem 1 is a simplification of the superization processes taking place in the successive layers of saliency maps. We have both class formation and the compound formation superization, and the computed entropy is spatial. We calculate superizations at the level of saliency maps. In other words, our signs and supersigns refer to values computed in successive saliency maps computed by the Grad-CAM method.

Our hypothesis is that, at the core of a CNN, both types of superizations exist. For type I superization (by class formation), the pooling operation combines signs (scalar values) by criteria like average value or maximum value, resulting in a single sign, and thus reducing their number and building equivalence classes. Another potential interpretation of the pooling operation is that it builds equivalence classes by grouping spatially neighboring elements. In our experiments (as we will see in Section 6), this phenomenon could be noticed after each pooling layer, where the magnitude of the spatial entropy of the saliency maps would have a big drop. Visually, the saliency maps start to become more concentrated around connected regions as more complex signs are formed.

For type II superization (by compound formation), it is known that CNNs compose whole objects starting from simple object parts [9]. This phenomenon describes exactly the second type of superization, as it builds compound supersigns from simpler component supersigns. They manage to do so by gradually enlarging the receptive field after each convolutional layer is applied. As the receptive field grows, a single neuron inside a hidden layer can cover a much larger region of interest from the input image and thus get activated for more and more complex objects.

What complicates the interpretation in case of CNN networks is the fact that for some layers both superizations operate simultaneously, and it can be difficult to separate their effects.

Our hypothesis is that, in order to decrease the spatial entropy noticeably, the first type of superization is more effective, while the second type is more responsible with building supersigns with semantic roles, not affecting spatial entropy that much.

## 6. Experiments

The goal for the next experiments is to explore the variation of the spatial entropy of the saliency maps computed with Grad-CAM on some representative CNN architectures. We expect the entropy to decrease along with depth, and this can be related to type I superization processes.

We consider three standard network architectures: AlexNet [2], VGG16 [3], and ResNet50 [5]. In addition, we also study the entropy variation on a custom LeNet-5-like network (The original LeNet-5 was introduced in [1]).

We use the deep learning programming framework PyTorch [42] (version 1.4.0) and the public implementation of Grad-CAM (https://github.com/utkuozbulak/pytorch-cnn-visualizations), modified to our needs. Except the custom network, all CNNs are used as provided by the PyTorch repository, with their default pretrained weights.

The experiments are performed in different contexts on the following datasets:A subset of ImageNet [43] composed of the “beaver” class from the training set, to test the pretrained and randomly initialized use-cases.CIFAR-10 [44] to: *(a)* train the custom network without downsampling; and *(b)* test the newly trained network and a randomly initialized one, with the same architecture, using this dataset as a test set.“kangaroo” class from Caltech101 [45] to test a network pretrained on ImageNet. The fact that we train and test on different (but somehow similar) datasets can have an impact on the generalization performance of the network and expose possible overfitting on the training data. This is known as *zero-shot* learning, and it can be viewed as an extreme case of domain adaptation.Caltech101 [45] to test for the case where the network is pretrained on ImageNet, then trained (fine-tuned) on Caltech101. This is the *transfer learning* approach.

### 6.1. Experiments on Standard CNN Architectures

We present the experimental results for each of the considered CNN architectures. In the next tables, we use the following terms: (i) Pretrained—publicly available pretrained weights on ImageNet, (ii) Random—randomly initialized weights, (iii) Fine-tuning—fine-tuned weights starting from the pretrained ones trained on ImageNet, (iv) ImageNet—“beaver” class from the ImageNet training set, (v) Caltech101—“kangaroo” class from the Caltech101 training set.

AlexNet [2] is composed of a sequence of convolutional, max-pooling, and ReLU layers, followed at the end by fully connected layers which linearly project the extracted features from the convolutional backbone to the desired number of output classes. Table 1 captures the experimental result values for each layer of the network.

VGG16 [3] has a relatively simple and compact architecture, consisting of only 3×3 convolutions, max-pooling, and ReLU, followed by multiple fully connected layers. The trick behind the VGG16 architecture is to use two 3×3 sequential convolution to replace a bigger 5×5 one, thus obtaining the same receptive field coverage by using less parameters. The caveat of VGG16 is that most of its parameters reside in the fully connected layers, making the network very parameter and memory inefficient. Table 2 depicts the entropy values at different levels of the network.

The novelty of ResNet [5] stands in the residual connections which alleviate the vanishing gradient problem, an issue that followed deep neural networks since their early days. During backpropagation, gradients would start to gradually decrease in magnitude because of the chain rule applied to very small values, until they become 0, and, consequently, many layers would lack any gradient signal on which basis to update their respective weights. ResNet solves this problem by creating residual branches from an input block to an output block in the form of y=x+f(x), where *x* is the block’s input and f(x) is a sequence of multiple layers. Instead of learning a function, as in earlier architectures like AlexNet or VGG16, ResNets are trying to learn a residual for the input *x*, hence the name of the architecture. Entropy values for various layers are shown in Table 3.

For all three networks, we observe a tendency of the spatial entropy to decrease, especially after max-pooling layers, which in our hypothesis are layers responsible for type I superization. Type II superization can be noticed by applying multiple consecutive convolutional layers. In this case, the spatial entropy does not necessarily decrease, but the general purpose is to enlarge the receptive field of the network, such that neurons activate for more complex objects while progressing through the layers.

Considering our above experiments and the well known fact that CNNs compose complex objects starting from simpler ones, this supports our hypothesis that type I superization is more effective for the entropy decrease. We did not notice a systematic entropy decrease for type II superization, and conclude that it is more responsible for building supersigns with semantic roles.

To prove the benefic effects of transfer learning when fine-tuning, we also train starting from a random initialization. We use the Caltech101 dataset [45], since it consists of real images like the ones in ImageNet. The results for fine-tuning and training from scratch are available in Table 4. For both experiments, we use a learning rate of 0.001 and train the networks for 100 epochs. The training set consists of the full dataset, apart from five random samples for each class, which are held for testing. The results when training from scratch are clearly worse than when fine-tuning from a strong baseline. Since the new training dataset is very small (≈9000 samples) compared to the size of ImagetNet (≈1.3 M samples), the network overfits on the training samples. This explains the weak performance when training from scratch.

### 6.2. Experiments on a Custom Network

Since all standard CNNs use a form of downsampling, either through strided convolutions or pooling, we notice that type I superization is always present. In these standard CNNs, both superization types are simultaneously present. The question is how to isolate the type II superization from the type I superization.

For this, we create a custom network by removing all spatial subsampling operations (strided convolutions and max-poolings) from original LeNet-5. This way, we remove the type I superization (class formation) and analyze entropy variation with respect to type II superization (compound formation) only.

We add two more convolutional layers to increase the receptive field of the network such that it can build more complex type II supersigns and simply have more layers to study the spatial entropy. The architectural details are depicted in Table 5.

We train this network on CIFAR-10 for 20 epochs, with the Stochastic Gradient Descent (SGD) optimizer and a learning rate of 0.01, until it reaches ≈72% accuracy on the test set, and then use it to generate the saliency maps. The accuracy performance is less relevant, since in this experiment we focus on the variation of the entropy. In Table 6, we can observe that the entropy does not vary too much for both the pretrained and random versions, but the random one exhibits much larger values.

## 7. CNN Architecture Optimization

It is known that modern neural network architectures are overparametrized [46], and so, an important emerging trend in deep learning is the optimization of such deep neural networks to satisfy various hardware constraints. An overview of such optimization techniques can be found in [47,48]. Among them, pruning is regarded as a fundamental method which has been studied since the late 1980s [49], and consists of reducing redundant operations by means of removing unnecessary or weak connections at the level of weights or layers. In the last couple of years, the state-of-the-art pruning methods have advanced considerably and are now capable of reducing the computational overhead of a deep neural network by a few times without incurring any loss in accuracy [50].

The experiments described in Section 6 showed that the spatial entropy of the CNN saliency maps generally decreases layer by layer, and we can relate this to semiotic superization. We aim to show how this interpretation could also help to optimize (or simplify) the architecture of the network. We perform an ablation study to see if we can determine redundant layers for pruning based on the spatial entropy information of the saliency maps. It is beyond the scope of this paper to systematically compare our approach with other CNN architecture optimization techniques. We only explore this area as a proof-of-concept, since it is the first time that such a semiotic method is used for neural architecture optimization.

On the VGG16 network, we iteratively apply the following greedy algorithm: (i) train the network on CIFAR-10 using the SGD optimizer with a learning rate of 0.01; (ii) compute the spatial entropy for each saliency map; (iii) remove a layer for which the entropy does not decrease; and (iv) repeat steps (i)–(iii) until the performance does not degrade too much.

From the results (see Figure 2), we notice that up to eight convolutional layers can be completely removed from the network, and this affects the performance by less than 1%. When removing the 9th layer, the accuracy decreases significantly; therefore, we stop the iterative process at this stage.

An interesting finding is that the order in which we remove layers matters significantly. If small layers with few parameters from the beginning of the network are removed first, the accuracy goes down by 2% after the 3rd removal. When removing from the big (over-parametrized) layers starting from the mid-end level of the network, the accuracy is maintained. The accuracy degrades especially fast after the 2nd convolutional layer with 64 output channels being removed.

Our explanation is that the first two convolutional layers are crucial for the downstream performance of the network. This first part of a network, before a subsampling operation is applied, is known in the literature as stem [51]. Some variants of ResNets implement this stem as three 3×3 convolutional layers or a big 7×7 layer. These early layers are responsible with detecting low level features like edge detectors. Having only a 3×3 convolutional layer, instead of two or three, means that the receptive field before the first max-pooling operation is 3×3, which might be too small to properly detect basic strokes and edges.

The resulted network has the following configuration: 64, 64, M, 128, M, 256, M, 512, M, M, where “M” stands for max-pooling and the integers represent a convolutional layer with the respective number of output channels, followed by a ReLU nonlinearity. The fully connected layers do not change from the original architecture. We compare our resulted network with VGG11, which is the smallest architecture from the VGG family. The results are displayed in Table 7. It can be noticed that, even when reducing the network capacity by a factor of approximately 7.5×, the accuracy is still maintained, meaning that the network is too over-parametrized for this task.

To check that the configuration translates to other tasks as well, we also trained the network on CIFAR-100 and compared it with the full VGG16’s performance. For the full VGG16 network, we obtain an accuracy of 62.61%, whereas for the optimized VGG architecture we get 63.78%. As can be seen, the small network even slightly improves the performance of the full network, while being much smaller.

We can visualize (Figure 3) this iterative removal experiment by plotting the saliency maps for a CIFAR-10 image from the “truck” class at different key layers, where the spatial entropy value saw a large drop from the previous layer, starting from the full VGG16 network and removing one layer at a time. The rows in Figure 3 represent layers at a particular depth, while the columns different architecture configurations found by the iterative method described above. In compliance with the theory of semiotic superization, it is visible how supersigns are gradually formed, layer by layer, from simpler supersigns. We observe this phenomenon from the fact that yellow regions (which denote pixel importance) become more structured and connected as we traverse through the layers. If we compare the saliency maps from the first column (corresponding to the full network) to the ones preceding them, we notice that the overall structure is maintained across all architecture configurations. This suggests that semiotic superization takes place inside a deep neural network regardless of the architecture of the network.

## 8. Conclusions

We introduced a novel computational semiotics interpretation of CNN architectures based on the statistical aspects of the information concentration processes (semiotic superizations) which appear in the saliency maps of successive CNN layers. At the core of a CNN, the two types of superization co-exist. According to our results, the first type of superization is effective at decreasing the spatial entropy. Type II superization is more responsible for building supersigns with semantic roles.

Beyond the exploratory aspect of our work, our main insights are twofold. On the knowledge extraction side, the obtained interpretation can be used to visualize and explain decision processes within CNN models. On the neural model optimization side, the question is how to use the semiotic information extracted from saliency maps to optimize the architecture of the CNN. We were able to significantly simplify the architecture of a CNN employing a semiotic greedy technique. While this optimization process can be slow, our work tries to use the notion of computational semiotics to prune an existing state-of-the-art network in a top-down approach instead of constructing one using a bottom-up approach like neural architecture search. Thorough analysis has to be done in future work to consider other network architectures and robustness of the method.

Some computational improvements for calculating the spatial entropy were proposed by Razlighi et al. [52,53]. The computational overhead can be significantly reduced if we accept a reduction of the approximation accuracy. We plan to use this trick in the future.

In this work, we considered only one type of neural network topology: CNNs. Since CNNs are mostly suited for images, those became the subject of our study. In the future, we intend to study the connection with other fields (audio, text) and architecture types (recurrent neural networks). The semiotic approach can be extended to other deep learning models, since semiotic superization appears to be present in many architectures. The computational semiotics approach is very promising especially for the explanation and optimization of deep networks, where multiple levels of superization are implied.

## Figures and Tables

**Figure 1 entropy-22-01365-f001:**
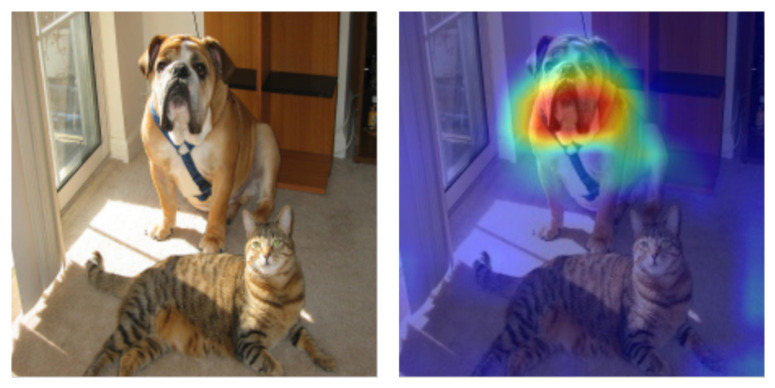
A saliency map (generated using the Grad-CAM method) which highlights the most important pixels that contribute to the prediction of the class “boxer” (dog). Red denotes important regions.

**Figure 2 entropy-22-01365-f002:**
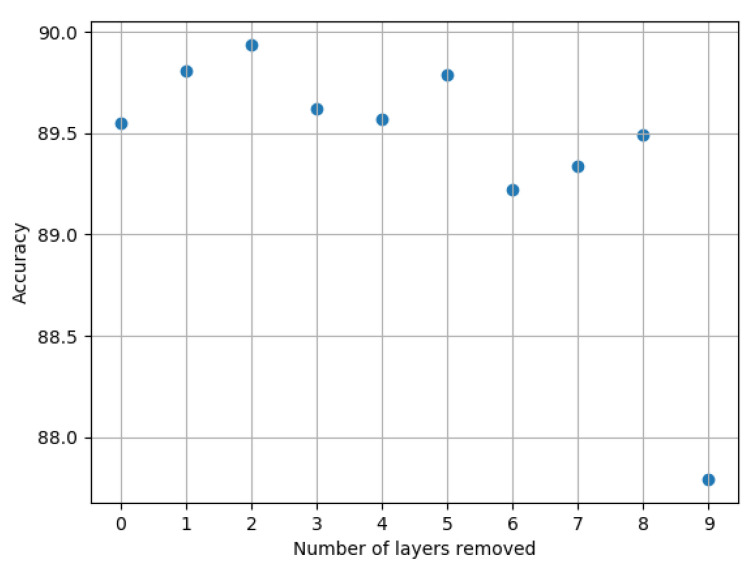
Accuracy measurements of VGG16 on CIFAR-10 as more layers are removed.

**Figure 3 entropy-22-01365-f003:**
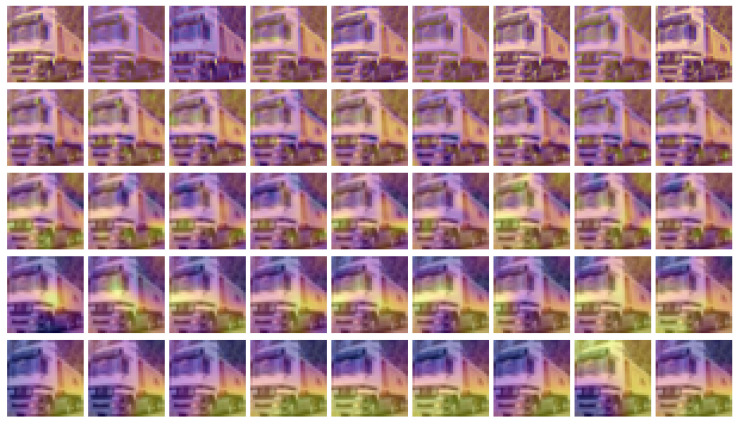
Saliency maps of a truck image from the CIFAR-10 dataset. Each row represents a layer within the CNN. Each column represents a new network configuration from which we removed one layer from the previous structure, as detailed above. We used the “plasma” effect to indicate the hotness of the saliency map, where the transition from purple to yellow denotes more important regions.

**Table 1 entropy-22-01365-t001:** Entropy values for saliency maps for AlexNet at different levels in the network.

AlexNet				
**Layer**	**Pretrained**	**Random**	**Pretrained**	**Fine-tuning**
	**ImageNet**	**ImageNet**	**Caltech101**	**Caltech101**
conv1	0.6830	0.6816	0.6786	0.6829
relu1	0.6806	0.6802	0.6746	0.6795
maxpool1	0.5252	0.5113	0.5264	0.5356
conv2	0.5311	0.5100	0.5395	0.5352
relu2	0.5231	0.5096	0.5297	0.5191
maxpool2	0.4147	0.3952	0.4241	0.4116
conv3	0.4423	0.3861	0.4508	0.4474
relu3	0.4326	0.3864	0.4437	0.4454
conv4	0.4272	0.3867	0.4375	0.4292
relu4	0.4214	0.3934	0.4222	0.4304
conv5	0.4056	0.3934	0.4019	0.3925
relu5	0.3928	0.3949	0.3878	0.3784
maxpool3	0.3114	0.3038	0.3077	0.3071

**Table 2 entropy-22-01365-t002:** Entropy values for saliency maps for VGG16 at different levels in the network.

VGG16				
**Layer**	**Pretrained**	**Random**	**Pretrained**	**Fine-tuning**
	**ImageNet**	**ImageNet**	**Caltech101**	**Caltech101**
conv1	0.8516	0.785	0.8418	0.8369
conv3	0.8017	0.7322	0.7883	0.7731
conv5	0.6742	0.6308	0.6681	0.648
conv10	0.5491	0.5155	0.5556	0.5429
conv12	0.5112	0.5155	0.5127	0.4901
conv13	0.4213	0.4035	0.4281	0.4135
conv14	0.3868	0.4288	0.3994	0.3599
maxpool5	0.3131	0.3443	0.3238	0.3086

**Table 3 entropy-22-01365-t003:** Entropy values for saliency maps for ResNet50 at different levels in the network.

ResNet50				
**Layer**	**Pretrained**	**Random**	**Pretrained**	**Fine-Tuning**
	**ImageNet**	**ImageNet**	**Caltech101**	**Caltech101**
conv1	0.7854	0.6574	0.7705	0.7633
block1	0.6849	0.5108	0.6807	0.6794
block2	0.5912	0.4193	0.5901	0.582
block3	0.4574	0.3398	0.4588	0.4607
block4	0.2847	0.3019	0.2754	0.2862

**Table 4 entropy-22-01365-t004:** Experimental results for accuracy on the Caltech101 dataset when fine-tuning from the available ImageNet pretrained weights versus starting from scratch.

Network	Fine-Tuning from ImageNet	Training from Scratch
AlexNet	83.168%	42.376%
VGG16	87.327%	61.584%
ResNet50	92.673%	43.168%

**Table 5 entropy-22-01365-t005:** Custom network architecture for the CIFAR-10 use-case.

Custom Network			
**Layer**	**Kernel Size**	**Input Channels**	**Output Channels**
Conv1 + ReLU	3×3	3	6
Conv2 + ReLU	3×3	6	16
Conv3 + ReLU	3×3	16	24
Conv4 + ReLU	3×3	24	32
Fc1 + ReLU	-	32×32×32	120
Fc2 + ReLU	-	120	84
Fc3	-	84	10

**Table 6 entropy-22-01365-t006:** Entropy values for saliency maps for the custom network at different levels in the network.

Custom Network with No Spatial Downsampling		
**Layer**	**Pretrained**	**Random**
conv1	0.4273	0.5948
relu1	0.4454	0.6062
conv2	0.4505	0.5802
relu2	0.5025	0.6189
conv3	0.4354	0.6101
relu3	0.4661	0.6123
conv4	0.4187	0.5674
relu4	0.4415	0.5798

**Table 7 entropy-22-01365-t007:** Comparisons on CIFAR-10—top 1 accuracy between VGG16, VGG11 (the smallest configuration from the VGG family), VGG16 after four layers removed (which has roughly the same number of parameters as VGG11) and VGG16 after eight layers removed (which is the smallest configuration which maintains the accuracy within a 1% difference).

Network	Number of Parameters	Accuracy
VGG16	15,245,130	89.55%
VGG11	9,750,922	87.83%
VGG16		
after 4 layers	9,345,354	89.57%
removed		
VGG16		
after 8 layers	2,118,346	89.49%
removed

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
