# Peer review of "Semiotic Aggregation in Deep Learning"

_entropy, 2020, doi:10.3390/e22121365_

Round 1
Reviewer 1 Report
General Overview:
The authors propose a method to evaluate how Convolutional Neural
Networks interpret an image to build saliency maps. They use an
entropy function and the Grad-CAM technique to describe the semiotic
level of complexity "understood" by the different layers of a CNN.
The idea is interesting and worth pursuing, however, I believe that
there are a number of issues that must be resolved before publishing
the manuscript. These issues are important to support the
conclusions. The article is well written, duly and truthfully
supported by scientific literature.
The main comments we do originate in an unsatisfactory discussion on
the definition of the two types of superization, in the context of
this manuscript. We don't question considerably the conclusions,
but we ask for clearer and deeper explanations.
Specific points:
Typo in key words: “semiotics”
1. Introduction
The introduction is well written and clear.
2. Saliency Maps in CNNs
The concept of “saliency map” is widely and clearly explained, but
there is not a clear definition of the term “activation map” (lines
89,122) which may be confusing in section 2. An accurate
understanding of the Grad-CAM technique is relevant to follow the
manuscript so that more details on the formalism are needed. For
example what is understood for the RELU activation functions or how it is
interpreted. As a suggestion, the text could be reordered with the
last paragraph of the next section bringing a starting point to
answer this.
3. Image Spatial Entropy
Well explained.
4. Semiotic Aggregation in Deep Learning
The written language example given to describe the two superization
types supports the definitions well. But a concrete example that
distinguishes these two superization types in the context of image
interpretation would be more accurate and helpful in the context of
this manuscript. (lines 203 to 210)
(line 271-272) “the relation between” is repeated.
5. Signs and Supersigns in CNN Saliency Maps
Please, provide a clearer explanation on how the two types of
superization are distinguished and detected in the analysis. For the
first type it is argued that it can be detected by drops of the
entropy function defined above, but in the second type we would
appreciate a wider discussion.
6. Experiments
The explanations on how to proceed to perform the experiments are
good and well structured. Please, provide a clearer explanation on
how the two types of superization are detected by this analysis.
(lines 348-535) How an increase of the spatial entropy can be
interpreted as a type II superization? Please justify more in depth
how the interpretation of these results do confirm your
hypothesis. Similarly, (line 372) Please discuss how removing spatial
subsampling removes type I superization.
7. CNN architecture optimization
The manuscript shows that CNN architecture could be very significantly reduced
with a very marginal impact on accuracy, the reduction from VGG16 to
VGG16 after 8 layers removed is particularly impressive.
This is managed by removing layers from trained CNNs, simplifying them
through a top-bottom construction. In order to obtain the final
structure, one must in addition train the big network before having
the short one. Please explain in which sense this could be a benefit
for the Machine Learning community, or how could this be obtained from a
bottom to top construction (directly trained as a small structure).
Figure 2: The impact of removing n layers depends on which layers are
removed. How is the accuracy affected if we try to remove other
layers? Is the jump from removing 8 to 9 layers maintained with any
other configuration? How robust is this conclusion? Please address these
questions.
8. Conclusions
In section 4, semiotics is described with a very generic and general
definition and not at all limited to saliency maps in images. It is
argued that the conclusions of this survey are of interdisciplinary
interest, nevertheless the implications of the conclusions for fields
outside semiotics for image interpretation are not clear (e.g. how
this could be applied or interpreted on other semiotics themes like
language, sound).
Author Response
First of all, we'd like to thank you for the high quality of your review. It has been very helpful for us. In what follows you can find our corrections:
Typo in key words: “semiotics”
Corrected
- Saliency Maps in CNNs
The concept of “saliency map” is widely and clearly explained, but there is not a clear definition of the term “activation map” (lines 89,122) which may be confusing in section 2.
Changed to feature map, because it is more commonly used.
Added explanation for what a feature map is:
“feature map (i.e. the output of a convolutional layer, a maxpool layer, a nonlinear activation function)”
An accurate understanding of the Grad-CAM technique is relevant to follow the manuscript so that more details on the formalism are needed. For example, what is understood for the RELU activation functions or how it is interpreted. As a suggestion, the text could be reordered with the last paragraph of the next section bringing a starting point to answer this.
Added:
“The ReLU activation function is applied because only features that have a positive influence on the class of interest usually matter. Negative values are features likely to belong to other categories in the image. In \cite{DBLP:journals/corr/SelvarajuDVCPB16} the authors justify that without the ReLU function, the saliency maps could sometimes highlight more than just the desired class of interest. ”
Moved from the next section:
“$$O_{Grad-CAM}^{(l)}$ in Formula (\ref{sal-output}) is the output resulted by applying the Grad-CAM technique on a particular layer $l$. By normalizing the values between $[0, 1]$ using the min-max normalization scheme and then multiplying by $255$ it will result a map of pixel intensities $g \in [0, 255]$, where $255$ denotes maximum importance and $0$ denotes no importance.”
Changed the final paragraph of the Image Spatial Entropy section with:
“Putting it all together, starting from a map obtained by Formula (\ref{sal-output}), we compute the probabilities $p_{gg'}$ in Formula (\ref{formula:2}), and finally the AME in Formula (\ref{eq:6}).”
- Semiotic Aggregation in Deep Learning
The written language example given to describe the two superization types supports the definitions well. But a concrete example that distinguishes these two superization types in the context of image
interpretation would be more accurate and helpful in the context of this manuscript. (lines 203 to 210)
We added in Section 5 (see below) an example from image interpretation. This was a very helpful suggestion.
(line 271-272) “the relation between” is repeated.
Done.
- Signs and Supersignsin CNN Saliency Maps
Please, provide a clearer explanation on how the two types of superization are distinguished and detected in the analysis. For the first type it is argued that it can be detected by drops of the entropy function defined above, but in the second type we would appreciate a wider discussion.
We added after the Theorem the following:
“An intuitive application of this theorem is when we consider the neural layers of a CNN. A type I superization appears when we reduce the spatial resolution of a layer $k+1$ by subsampling layer $k$. This is similar to class formation because we reduce the variation of the input values (i.e., we reduce the number of signs). In CNNs, this is typically performed by a pooling operator. The pooling operator can be considered as a form of non-linear down-sampling which partitions the input image into a set of non-overlapping rectangles and, for each such sub-region, it computes its mean (average pooling) or max value (max pooling). The formula for max pooling applied to a feature map $F$ at layer $k$ and locations $(i, j)$ with a kernel of $2x2$ is:
\begin{equation}
O_{i, j}(F) = max (F_{i, j}, F_{i+1, j}, F_{i, j+1}, F_{i+1, j+1})
\end{equation}
A type II superization is produced when applying a convolutional operator to a neural layer $k$. As an effect, layer $k+1$ will focus on more complex objects, composed of objects already detected by layer $k$. The convolutional operator for a feature map $F$ at layer $k$ and pixel locations $(i, j)$ with a $3x3$ kernel $W$ has the following formula:
\begin{equation}
O_{i, j}(F) = \sum_{x=0}^2 \sum_{y=0}^2 F(i + x, i + y) W(x, y)
\end{equation}
The output $O$ of the convolutional operator is a linear combination of the input features and the learned kernel weights. Thus, a resulting neuron will be able to detect a combination of simpler object forming a more complex one, by composition of supersigns.”
- Experiments
The explanations on how to proceed to perform the experiments are good and well structured. Please, provide a clearer explanation on how the two types of superization are detected by this analysis.
Changed to:
“We expect the entropy to decrease along with depth and that this can be related to superization processes” to “We expect the entropy to decrease along with depth and that this can be related to type I superization processes”
(lines 348-535) How an increase of the spatial entropy can be interpreted as a type II superization? Please justify more in depth how the interpretation of these results do confirm your hypothesis. Similarly, (line 372) Please discuss how removing spatial subsampling removes type I superization.
We changed:
" Type II superization can be noticed after two consecutive convolutional layers, where the spatial entropy does not decrease too much and even increases in some cases. This confirms our hypothesis that type I superization is more effective for the entropy decrease, whereas type II is more responsible with building supersigns with semantic roles.”
with:
“For all three networks we observe a tendency of the spatial entropy to decrease, especially after max-pooling layers, which in our hypothesis are layers responsible for type I superization. Type II superization can be noticed by applying multiple consecutive convolutional layers. In this case, the spatial entropy does not necessarily decrease, but the general purpose is to enlarge the receptive field of the network, such that neurons activate for more complex objects while progressing through the layers.”
We changed:
“This confirms our hypothesis that type I superization is more effective for the entropy decrease, whereas type II is more responsible with building supersigns with semantic roles.”
with:
“Considering our above experiments and the well-known fact that CNNs compose complex objects starting from simpler ones, this supports our hypothesis that type I superization is more effective for the entropy decrease, whereas type II is more responsible with building supersigns with semantic roles.”
- CNN architecture optimization
This is managed by removing layers from trained CNNs, simplifying them through a top-bottom construction. In order to obtain the final structure, one must in addition train the big network before having the short one. Please explain in which sense this could be a benefit for the Machine Learning community, or how could this be obtained from a bottom to top construction (directly trained as a small structure).
Added to Conclusions:
“While this optimization process can be slow, our work tries to use the notion of computational semiotics to prune an existing state of the art network in a top-down approach instead of constructing one using a bottom-up approach like neural architecture search.”
Figure 2: The impact of removing n layers depends on which layers are removed. How is the accuracy affected if we try to remove other layers? Is the jump from removing 8 to 9 layers maintained with any
other configuration? How robust is this conclusion? Please address these questions.
We mention the following aspect in the text:
Throughout our experiments we noticed that it is important to remove layers from the mid-end part of the network where most of the redundancy seems to be happening.
Answer:
This part is only meant as a proof of concept to see if the notion of spatial entropy can be used at all for network optimization and thorough analysis must be done in future work.
Added to Conclusions:
“Thorough analysis has to be done in future work to consider other network architectures and robustness of the method.”
- Conclusions
In section 4, semiotics is described with a very generic and general definition and not at all limited to saliency maps in images. It is argued that the conclusions of this survey are of interdisciplinary
interest, nevertheless the implications of the conclusions for fields outside semiotics for image interpretation are not clear (e.g. how this could be applied or interpreted on other semiotics themes like language, sound).
Added to the last paragraph in Conclusions:
“In this work we considered only one type of neural network topology: CNNs. Since CNNs are mostly suited for images, those became the subject of our study. In the future, we intend to study the connection with other fields (audio, text) and architecture types (recurrent neural networks).”
Reviewer 2 Report
The authors introduce an approach to the analysis of deep learning called Semiotic Aggregation which refers to the identification of signs representing syntactic, semantic, and to a lesser extent pragmatics in the structured layers of neural networks. The analysis is thorough and original, though the structure of the paper could be improved to strengthen their claims regarding computational semiotics. The introduction to semiotics, saliency maps, and image spatial entropy gives the reader a good background in the technical methods.
The theoretical discussion is good but drifts toward discussions of consciousness and psychology that detract from the technical achievements. For instance, in line 212, the suggestion that “superization is a conscious perception process” should be removed, since it’s not supported by factual experiments in this paper. Given this paper’s focus regarding computational algorithms issues of “conscious perception” are out of scope.
While the principal experimental results, documented in section 6, are the reduction in spatial entropy at successive layers of the deep learning network, it is difficult to follow the connection between these measurements and the conceptual framework of semiotic aggregation. The connection is not qualitatively established until the example in section 7 shows the saliency maps as a function of deep learning layers and the number of layers removed. The paper would be much more persuasive if the experimental results in section 6 include similar saliency maps so that a connection can be established between the reduction in entropy and the topology of the resulting saliency maps. Likewise, a tighter connection between the spatial entropy, the saliency maps, and the classification performance would be helpful.
The writing style of the document is very good. Some grammatical errors are noted. These and other detailed comments are provided as comments within the pdf document.

Author Response
First of all, we'd like to thank you for the high quality of your review. It has been very helpful for us. In what follows you can find our corrections:
The theoretical discussion is good but drifts toward discussions of consciousness and psychology that detract from the technical achievements. For instance, in line 212, the suggestion that “superization is a conscious perception process” should be removed, since it’s not supported by factual experiments in this paper.  Given this paper’s focus regarding computational algorithms issues of “conscious perception” are out of scope.
We agree: we have modified this (see similar observation below).
While the principal experimental results, documented in section 6, are the reduction in spatial entropy at successive layers of the deep learning network, it is difficult to follow the connection between these measurements and the conceptual framework of semiotic aggregation.  The connection is not qualitatively established until the example in section 7 shows the saliency maps as a function of deep learning layers and the number of layers removed. The paper would be much more persuasive if the experimental results in section 6 include similar saliency maps so that a connection can be established between the reduction in entropy and the topology of the resulting saliency maps.
Answer:
We did not include any saliency maps in Section 6 because the ones in Section 7 fully describe the visual behavior for saliency maps in general, for any network. Early layers will tend to have more discontinuous regions, while the latter ones will be more concentrated around a single connected region.
Likewise, a tighter connection between the spatial entropy, the saliency maps, and the classification performance would be helpful.
Answer:
The classification performance is independent of the spatial entropy or saliency maps, and usually depends on the capacity/complexity of the network being used. We only use the spatial entropy as a heuristic to choose which layers to prune, and this is already mentioned in the paper.
According to the attached PDF observations, we did the following:
Rewrite this to be clear that the rows are samples from different depths of the network and the columns represent removal of layers.
Added:
“The rows in Figure \ref{fig:saliency_removal} represent layers at a particular depth, while the columns different architecture configurations found by the iterative method described above.”
This section needs to be expanded upon since it is the key insight of the paper. Since yellow indicates importance, there appears to be an increase in the identification of important regions by removing layers. For instance, the bottom image second from the right has a very bright yellow region at the top of the cab. How does the entropy and accuracy of images in this configuration compare with the neighboring configurations?
Changed:
“we notice that the overall structure is maintained across all levels of layer removals” with “we notice that the overall structure is maintained across all architecture configurations”
Answer:
The increase in the identification of important regions is due to the depth of the layer and not the removal of a particular one.
the bottom image second from the right has a very bright yellow region at the top of the cab
Answer:
In this instance the Grad-CAM technique decided that the most salient region for classifying that truck was the top part of the image. The spatial entropy value for that saliency map is similar to the ones corresponding to different architectures (columns) at the same depth (last row).
This section needs to be expanded upon since it is the key insight of the paper
Answer:
The main scope of this qualitative experiment was to prove how supersigns are built by visually looking at saliency map connectivity.
Move this sentence closer to the beginning of the introduction and use the phrase "We provide an original, and to our knowledge, the first application ...". It will be important that the paper clearly defines "computational semiotics" and demonstrates why their analysis of deep neural networks is truly distinct and original in comparison to other methods. Otherwise, the claim of "first" may not be valid.
We added the following paragraph in the Introduction:
“Our contribution is an original, and to our knowledge, the first application application of computational semiotics in the analysis and interpretation of deep neural networks. \emph{Semiotics} is known as the study of signs and sign-using behavior. According to \cite{Gudwin2005a}, \emph{computational semiotics} is an interdisciplinary field which proposes a new kind of approach to intelligent systems, where an explicit account for the notion of sign is prominent. In our work, the definition of computational semiotics refers to the application of semiotics to artificial intelligence. We put the notion of sign from semiotics into service to give a new interpretation of deep learning, and this is new. We use computational semiotics' concepts to explain decision processes in CNN models. We also study the possibility of applying semiotic tools to optimize the architecture of deep learning neural networks. Currently, model architecture optimization is a hot research topic in machine learning.”
We tried to move the above paragraph closer to the beginning of Introduction but failed. This would destroy the flow of ideas. It is better (in our opinion) to have the following sequence: context, motivation, contribution, structure of the paper.
Like the use of 'consciousness' below, this phrase seems over-wrought. What does this phrase mean? Conscious implies intentionality; however, the focus is on a mechanical process regarding multilayer processing of an image. Defining and specifying consciousness is an open scientific problem. It cannot at this time be assigned to a computer process. This type of language should be removed from the paper. It's a distraction from documenting the performance of an engineering result.
Answer:
We agree – it is a distraction, even if it is an interesting point of view of the cited author. We have removed these sentences.
Since a theorem and its proof should be able to read independently of the rest of the text, add a brief synopsis of the two types of superization.
We added after the Theorem the following:
“An intuitive application of this theorem is when we consider the neural layers of a CNN. A type I superization appears when we reduce the spatial resolution of a layer $k+1$ by subsampling layer $k$. This is similar to class formation because we reduce the variation of the input values (i.e., we reduce the number of signs). In CNNs, this is typically performed by a pooling operator. The pooling operator can be considered as a form of non-linear down-sampling which partitions the input image into a set of non-overlapping rectangles and, for each such sub-region, it computes its mean (average pooling) or max value (max pooling). The formula for max pooling applied to a feature map $F$ at layer $k$ and locations $(i, j)$ with a kernel of $2x2$ is:
\begin{equation}
O_{i, j}(F) = max (F_{i, j}, F_{i+1, j}, F_{i, j+1}, F_{i+1, j+1})
\end{equation}
A type II superization is produced when applying a convolutional operator to a neural layer $k$. As an effect, layer $k+1$ will focus on more complex objects, composed of objects already detected by layer $k$. The convolutional operator for a feature map $F$ at layer $k$ and pixel locations $(i, j)$ with a $3x3$ kernel $W$ has the following formula:
\begin{equation}
O_{i, j}(F) = \sum_{x=0}^2 \sum_{y=0}^2 F(i + x, i + y) W(x, y)
\end{equation}
The output $O$ of the convolutional operator is a linear combination of the input features and the learned kernel weights. Thus, a resulting neuron will be able to detect a combination of simpler object forming a more complex one, by composition of supersigns.”
class formation and the compound formation” This specification is necessary since its a new section
Done.
More effective at what?
Modified to:
“According to our results, the first type of superization is more effective at decreasing the spatial entropy, while the second type is more responsible with building supersigns with semantic roles.”
Reviewer 3 Report
Summary: The purpose of this manuscript is to analyze the saliency maps of information concentration in successive layers using computational semiotics. A decrease of spatial entropy and increase of information entropy is used in the aggregate operation. The information content of the saliency maps is calculated by using spatial entropy. The superization processes is visualized. The results are used to explain the neural decision model ...
Evaluation: The manuscript is well written and easy to understand. The context of the research subject is interesting. The organization of this manuscript is good, and the material is understandable. The links to the references is good.
Author Response
First of all, we'd like to thank you for the high quality of your review. It has been very helpful for us. In what follows you can find our corrections:
It is suggested that the authors edit the manuscript because of English errors.
We have made some small language corrections also suggested by the other reviewers
It is suggested that the authors explain more in details about the contribution of this paper.
We added the following paragraph in the Introduction:
“Our contribution is an original, and to our knowledge, the first application application of computational semiotics in the analysis and interpretation of deep neural networks. \emph{Semiotics} is known as the study of signs and sign-using behavior. According to \cite{Gudwin2005a}, \emph{computational semiotics} is an interdisciplinary field which proposes a new kind of approach to intelligent systems, where an explicit account for the notion of sign is prominent. In our work, the definition of computational semiotics refers to the application of semiotics to artificial intelligence. We put the notion of sign from semiotics into service to give a new interpretation of deep learning, and this is new. We use computational semiotics' concepts to explain decision processes in CNN models. We also study the possibility of applying semiotic tools to optimize the architecture of deep learning neural networks. Currently, model architecture optimization is a hot research topic in machine learning.”
Round 2
Reviewer 1 Report
General Overview:
The authors propose a method to evaluate how Convolutional Neural
Networks interpret an image to build saliency maps. They use an
entropy function and the Grad-CAM technique to describe the semiotic
level of complexity "understood" by the different layers of a CNN.
The authors took into account many of our comments. Now, the manuscript is
clearer, in particular the two types of superizations are explained in
a more natural way.
Nevertheless we still detect some problems in this survey. It seems
that the proposed method does not detect type II of superization, thus
there is a need to resolve that (see 6. Experiments)
Particular comments:
2 Saliency maps
The short statement on feature maps (lines 97-98) has helped improve
the reading, as well as reordering the sections. Similarly, the description of the GRAD-CAM
technique is clearer now.
4. Semiotic Aggregation in Deep Learning
The relationship between the first class of superization is finally
observed in the max pooling or average pooling operations while the
second type in the convolutional operator. This makes perfect sense
with the definition given previously and favors significantly the
understanding of the next sections.
5. Signs and Supersigns in CNN Saliency Maps
Changes made in the previous section provide a satisfactory answer.
6. Experiments
The detection of type I superization is clearer now: the entropy drops
down. Nevertheless it seems that type II has basically no effect on
the entropy, so how is it detected? It is argued that type II “is more
responsible for building supersigns with semantic roles”, no doubt it
is true since the convolutional layers of the CNN are designed to do
so, but no measure of this effect seems to be provided by this method.
line 383 should replace: “more responsible with” by “more responsible
for”
7. CNN architecture optimization
Changes are satisfactory.
8. Conclusions
If this method can only be used for type I, then the conclusions have
to be restated accordingly